# Does the LDH/Albumin Ratio Bring Novelty? A Comparative Analysis with Inflammatory Indices and Combined Models in Adult-Onset Still’s Disease

**DOI:** 10.3390/diagnostics14242780

**Published:** 2024-12-11

**Authors:** Ali Ekin, Salim Mısırcı, Hikmet Öztop, Asuman Şebnem Hacımustafaoğlu, Belkıs Nihan Coşkun, Burcu Yağız, Ediz Dalkılıç, Yavuz Pehlivan

**Affiliations:** 1Divison of Rheumatology, Department of Internal Medicine, Faculty of Medicine, Bursa Uludag University, 16059 Bursa, Turkey; dr.salim-misirci@hotmail.com (S.M.); belkisnihanseniz@hotmail.com (B.N.C.); burcuyilmaz_84@hotmail.com (B.Y.); edizinci@hotmail.com (E.D.); drypehlivan@gmail.com (Y.P.); 2Department of Internal Medicine, Faculty of Medicine, Bursa Uludag University, 16059 Bursa, Turkey; hikmetoztop@gmail.com; 3Gaziantep City Hospital, 27470 Gaziantep, Turkey; sbnm.hacimustafaoglu@gmail.com

**Keywords:** adult-onset Still’s disease, albumin, inflammation, fever of unknown origin, lactate dehydrogenase/albumin, LDH

## Abstract

Background/Objectives: The objective of this study was to evaluate the diagnostic accuracy of the lactate dehydrogenase-to-albumin ratio (LAR) in adult-onset Still’s disease (AOSD) and compare it with other inflammatory indices, using patients with fever of unknown origin (FUO) as a control group due to their overlapping clinical features with AOSD. The study also compared LAR’s diagnostic performance with other inflammatory indices like the serum immune-inflammatory index (SII), ferritin/erythrocyte sedimentation rate (FER), CRP/albumin ratio (CAR), platelet/lymphocyte ratio (PLR), and neutrophil/lymphocyte ratio (NLR), as well as its combinations with FER, PLR, and ferritin (LAR + FER, LAR + PLR, LAR + ferritin). Methods: A retrospective evaluation was conducted on 70 patients with fever of unknown cause and 78 patients with AOSD, admitted between January 2000 and December 2023 in a tertiary care hospital. Demographic, clinical, and laboratory characteristics were compared between the groups. ROC analysis provided cutoff values, sensitivity, and specificity for each inflammatory index. Results: ROC analysis showed significant *p*-values (*p* < 0.05) for indices other than LAR (*p* = 0.090) LAR + PLR (*p* = 0.806), and PLR (*p* = 0.634) in diagnosing AOSD. The highest specificity was found in LAR + ferritin (92.90%), and the highest sensitivity in CAR (100.0%). NLR, SII, FER, and LAR + FER were the indices with both sensitivity and specificity above 50%. LAR had a sensitivity of 76.90% and a specificity of 48.60%. The cutoff values were 3978.0 µg/L for ferritin and 70.98 for LAR. Significant statistical differences between AOSD and non-AOSD groups were observed for all indices except CAR (*p* = 0.133). Conclusions: LAR can differentiate AOSD patients from FUO, but its specificity is lower than most other indices. The diagnostic utility of these indices in clinical practice remains controversial.

## 1. Introduction

Adult-onset Still’s disease (AOSD), a rare systemic and autoinflammatory disease, is a complex disorder characterized by fever, salmon-colored rash, sore throat, arthritis, lymphadenopathy (LAP), organomegaly, and hyperferritinemia. The incidence of this disease, which usually affects young adults, is 0.16–0.4 per 100.000 [1,2]. Laboratory values are crucial for both the initial diagnosis and ongoing treatment of AOSD. Neutrophilic leukocytosis, an increase in liver enzymes such as aspartate aminotransferase (AST), alanine aminotransferase (ALT), gamma-glutamyl transferase (GGT), alkaline phosphatase (ALP), lactate dehydrogenase (LDH), erythrocyte sedimentation rate first hour (ESR), C-reactive protein (CRP), and marked increase in ferritin level are some of these parameters. However, these parameters, which indicate systemic inflammation in AOSD, are not specific for AOSD [1,3,4,5]. Although the Yamaguchi criteria are used for the diagnosis of AOSD, malignancy, infectious diseases and other rheumatologic diseases must be excluded [1].

The absence of a particular diagnostic tool for AOSD may be one of its biggest drawbacks, since it causes the diagnosis procedure to take longer than necessary [6]. Thrombotic thrombocytopenic purpura (TTP), macrophage activation syndrome (MAS), diffuse alveolar hemorrhage (DAH), and disseminated intravascular coagulation (DIC) are among the most lethal complications of AOSD with mortality rates between 10 and 41% [7]. It is crucial to diagnose and treat AOSD patients as soon as possible since the condition might be deadly for certain people. Data on the use of systemic immune inflammatory indices in rheumatologic diseases are increasing day by day [8]. The serum immune and inflammatory index (SII; platelet count ×neutrophil/lymphocyte count at diagnosis), the platelet/lymphocyte ratio (PLR), CRP/albumin ratio (CAR), the ferritin/erythrocyte sedimentation rate (FER), the neutrophil/lymphocyte ratio (NLR), and the LDH/albumin ratio (LAR) are inflammatory indices that have been used in many studies but are not widely used clinically [8,9,10,11,12].

LDH and albumin are easily accessible parameters that can be analyzed in many laboratories in a short time. The LAR, which is the result of dividing LDH by albumin, is related to systemic inflammation and malignant processes and has been used as an indicator for prognosis and prediction of infection in breast cancer, lymphoma, pancreatic cancer, esophageal cancer, colorectal cancer, lower respiratory tract infections, COVID-19 infections, acute ischemic stroke, and intensive care units [13,14,15,16,17,18]. When we reviewed the literature, there was one study that defined LAR in AOSD patients and suggested that its use could be predictive [10]. Another study compared the inflammatory indices SII, albumin/globulin ratio (AGR), CAR, SII+ ferritin, prognostic nutritional index (PNI), and FER in AOSD. No other study was found comparing LAR with FER, CAR, PLR, NLR, or SII [9]. When examining the data of inflammatory indices in the literature, one realizes that most of them are not useful at first sight and cannot be used in daily practice as much as ferritin. In fact, most studies, especially in malignant diseases, present the newly found or applied index as very ambitious and with significant positive likelihood ratio, negative likelihood ratio, specificity, and sensitivity data.

In our investigation, we were influenced by a prior study that examined the impact of LAR in AOSD without a control group and conducted a comparison within the same group based on ferritin levels [10]. We also drew inspiration from another study that evaluated the predictive value of SII + ferritin in AOSD with other markers of inflammation, utilizing a control group for comparison [9]. The objective of this study was to evaluate the diagnostic accuracy of the lactate dehydrogenase-to-albumin ratio (LAR) in adult-onset Still’s disease (AOSD) and compare it with other inflammatory indices, using patients with fever of unknown origin (FUO) as a control group due to their overlapping clinical features with AOSD. The study also compared LAR’s diagnostic performance with other inflammatory indices like the serum immune-inflammatory index (SII), ferritin/erythrocyte sedimentation rate (FER), CRP/albumin ratio (CAR), platelet/lymphocyte ratio (PLR), and neutrophil/lymphocyte ratio (NLR), as well as its combinations with FER, PLR, and ferritin (LAR + FER, LAR + PLR, LAR + ferritin). Additionally, we aimed to evaluate the practicality and usefulness of LAR and other immune inflammatory indices in routine clinical practice.

## 2. Materials and Methods

### 2.1. Study Groups and Patients

The research was based on a cohort of 182 individuals who received medical treatment at Bursa Uludag University’s medical faculty, which is a tertiary healthcare center. Out of the total number of patients, 102 individuals were identified as having the diagnostic code AOSD according to the International Classification of Disease (ICD) in the hospital’s medical data system. Out of the 102 patients who had a diagnostic code for AOSD, 20 patients were not included in the research because they were diagnosed with a different condition during the follow-up period, and 4 patients were not included because there were not enough data or some data were missing. Thus, 78 patients who were followed up with a diagnosis of AOSD at the Rheumatology Clinic of Bursa Uludağ University Faculty of Medicine between January 2000 and July 2023 were included in the study. These patients, who were included in the study by scanning the ICD code, were re-evaluated according to the Yamaguchi criteria. After excluding infections, malignancies, and other rheumatologic diagnoses, the patients who were classified as having AOSD according to the Yamaguchi criteria were included in the study [1].

Fever of unknown origin (FUO) was characterized by Petersdorf and Beeson in 1961 as a febrile sickness lasting longer than three weeks, with several body temperatures above 38.3 °C and an unclear diagnosis following a week-long hospital stay [19]. We re-evaluated 80 of 182 patients for fever of unknown origin (FUO), and 10 patients who were not diagnosed with FUO or had insufficient and/or missing data were excluded. The number of patients in the FUO group, defined below as the non-AOSD group, was 70. In our study, 78 patients from the AOSD group and 70 patients from the FUO group were compared (Figure 1). FUO was chosen as the control group because it represents a heterogeneous group of patients with persistent fever and clinical features overlapping with AOSD. This allowed us to assess the diagnostic performance of LAR and other inflammatory indices in distinguishing AOSD from similar clinical presentations. Patients were stratified into two groups—AOSD and FUO—for comparative analysis. The patients who were followed up for FUO had rheumatologic diagnoses such as RA, SLE, vasculitis, and malignancies and infections. The Clinical Research Ethics Committee of Bursa Uludag University Faculty of Medicine authorized this research on 21 February 2024, with the reference number 2024-2/6.

### 2.2. Variables and Procedures

The laboratory and clinical data of the patients were evaluated retrospectively using the medical data registration system. Age, gender, date of diagnosis, duration of disease, time from onset of symptoms to diagnosis, AOSD disease pattern (monocyclic, polycyclic or chronic articular pattern), coronary artery disease (CAD), diabetes mellitus (DM), chronic kidney disease (CKD), hypertension (HT), chronic obstructive pulmonary disease (COPD), asthma, hyperlipidemia (HL), comorbid conditions such as heart failure (HF), development of MAS, development of malignancy after diagnosis, clinical manifestations of the disease, diagnoses made based on investigations in patients followed up for FUO, treatments of patients with AOSD, laboratory values such as CRP, ESR, AST, ALT, GGT, ALP, blood count, blood lipid levels, fibrinogen, albumin, and LDH at the time of diagnosis were analyzed. LAR, NLR, PLR, CAR, FER, SII, LAR + FER, LAR + PLR, and LAR + ferritin were calculated and recorded. The reason for combining LAR only with PLR, FER, and ferritin was their high specificity and sensitivity rates. The initial modified Pouchet scores (fever, typical rash, arthritis, pleurisy, pericarditis, sore throat, pneumonia, myalgia, hepatomegaly, lymphadenopathy, or WBC > 15,000/µL, elevated liver function tests, ferritin > 3000 µg/L) were calculated to assess the disease activity of patients with AOSD [20].

### 2.3. Statistical Analysises

The statistical analyses were conducted using SPSS (version 28.0; IBM Corporation, Armonk, NY, USA). The mean ± standard deviation (minimum, maximum, median) was used to display normally distributed quantitative data, whereas non-normally distributed quantitative data was given as the median [percentile (P) 25, percentile (P) 75]. The presence of a statistically significant difference between the AOSD group and the non-AOSD group was assessed using Pearson’s χ^2^ test for categorical data, independent samples t-test for normally distributed numerical data, and the Mann–Whitney U-test for non-normally distributed numerical data. Receiver operating characteristic (ROC) curves for various inflammatory ratings were analyzed in order to identify the optimal cutoff values for distinguishing between individuals with and without AOSD. The Youden index, which is obtained by adding the sensitivity and specificity and subtracting 1, was used to establish the threshold values for the scores and parameters. The AUC (area under the curve) was used to assess the overall diagnostic efficacy. For each score, we computed the negative likelihood ratio (-LR), positive likelihood ratio (+LR), specificity, sensitivity, 95% confidence intervals (CI), and odds ratio (OR) using the optimum cutoff value. LR stands for the likelihood ratio, which measures the strength of evidence for diagnosing AOSD based on inflammatory levels. The formula for calculating the positive likelihood ratio (LR) is sensitivity divided by (1-specificity), whereas the formula for calculating the negative likelihood ratio (LR) is (1-sensitivity) divided by specificity. A Pearson correlation analysis was conducted to assess the associations between the disease activity index and the diagnostic scores. A Rho coefficient ranging from 0.250 to 0.500 suggests a weak correlation, while a range of 0.500 to 0.750 indicates a moderate connection, and a range of 0.750 to 1.000 suggests a strong correlation. A significance criterion of *p* < 0.05 was used for all statistical analyses.

## 3. Results

### 3.1. Clinical and Laboratory Features, Treatment, and Follow-Up of AOSD Patients

Of the 148 patients enrolled in the study, 78 were diagnosed with AOSD and 70 with non-AOSD. Patients with non-AOSD were screened for FUO and diagnosed with malignancy (37/70, 52.86%), infection (10/70, 14.29%), rheumatologic disease (6/10, 8.57%). Table 1 displays the basic clinical features of the patients and the comparison between the groups. Within the AOSD group, 53 patients (67.90%) were female and the mean age was 48.82 ± 15.41 years. The main clinical symptoms and findings in the AOSD group were arthralgia (91.0%), fever (89.70%), arthritis (78.20%), typical non-pruritic salmon-colored rash (74.40%), and sore throat (46.16%). Clinical findings differed between the two groups with the exception of those regarding myalgia (*p* = 0.058) and splenomegaly (*p* = 0.122). In terms of laboratory findings, AST, ALT, albumin, and CRP were higher in the non-AOSD group, while other parameters were higher in the AOSD group. In addition, AST, ALT, GGT, and ESR values were not statistically significantly different between the two groups. With the exception of LAR (*p* = 0.090), PLR (*p* = 0.634), and LAR + PLR (*p* = 0.806), the other inflammatory indices differed statistically significantly between the two groups. While CAR, PLR, and LAR + PLR were lower in the AOSD group, SII, NLR, FER, and LAR were higher. The prevalence of MAS was 12.80% in the AOSD group and 7.10% in the non-AOSD group examined for FUO, and there was no difference in the development of MAS between the two groups (*p* = 0.253).

In 78 patients with a median disease duration of 132 months, the mean time from symptom onset to diagnosis was 2.42 ± 7.85 months. Thirty-seven (47.40%) patients had a monocyclic AOSD pattern, 27 (37.60%) had a polycyclic AOSD pattern and 14 (17.90%) had a chronic articular pattern. All patients were taking steroids at the time of diagnosis, and 34 (43.60%) patients were receiving pulse glucocorticoids. During the follow-up period, methotrexate (84.60%) and hydroxychloroquine (61.50%) were the most commonly used conventional DMARDs, while tocilizumab (17.90%) and anakinra (17.90%) were the most commonly used biologic DMARDs (Table 2).

### 3.2. Relationship Between Inflammatory Indices and the Diagnosis of AOSD

The diagnostic utility of inflammatory indicators and ferritin for the diagnosis of AOSD was evaluated using ROC curve analysis. The ROC curve analysis yielded the following results: LAR had an AUC of 0.581 (95% CI, 0.485–0.676), SII had an AUC of 0.657 (95% CI, 0.569–0.744), NLR had an AUC of 0.609 (95% CI, 0.517–0.701), FER had an AUC of 0.477 (95% CI, 0.381–0.574), CAR had an AUC of 0.737 (95% CI, 0.657–0.817), and ferritin had an AUC of 0.732 (95% CI, 0.653–0.811). The results of the combinations LAR + PLR, LAR + FER, and LAR + ferritin were as follows, respectively: 0.488 (95% CI, 0.393–0.584), 0.696 (95% CI, 0.609–0.783) and 0.729 (95% CI, 0.649–0.808). FER had the most superior diagnostic performance, with ferritin, SII, NLR, LAR, PLR, CAR, LAR + ferritin, and LAR + FER following in descending order (Figure 2).

Sensitivity, specificity, positive LR, and negative LR values were calculated for each inflammatory index and ferritin using the Youden index. In the ROC analysis, the *p*-values for SII, NLR, FER, and CAR were statistically significant, except those for LAR (*p* = 0.090), PLR (*p* = 0.634), and LAR + PLR (*p* = 0.806). The indices with the highest specificity were FER (81.40%) and LAR + ferritin (92.90%), the indices with the highest sensitivity were CAR (100.0%) and LAR + PLR (*p* = 98.70%). NLR, SII, FER, and LAR + FER were the indices with both sensitivity and specificity above 50%. The sensitivity and specificity of LAR were 76.90% and 48.60%, respectively. The cutoff value for ferritin was 3978.0 µg/L for diagnostic performance in distinguishing AOSD and FUO groups. Using the cutoff values determined by ROC analysis, there was a significant statistical difference between AOSD and FUO groups for all indices except CAR (*p* = 0.133). For all indices, a higher number of patients were found in the AOSD group compared to the cutoff value (Table 3 and Table 4).

### 3.3. Examining the Relationship Between Inflammatory Indices and Some Factors That May Serve as Indicators of Disease Activity in the AOSD Group

The correlation between the modified Pouchet score, which is used for disease activity, the ferritin value, which has a very important place in the diagnosis and follow-up of AOSD, and the classic laboratory findings that indicate organ and system findings in the diagnosis and follow-up of AOSD, such as AST, ALT, hemoglobin, platelets, leukocytes, neutrophils, albumin, ESR, and CRP, as well as the inflammatory indices, was calculated using Pearson correlation. The association between LAR and ferritin was weakly positive (r = 0.461, *p* ≤ 0.001), as well as that with AST (r = 0.367, *p* ≤ 0.001). On the other hand, there was a faint negative correlation between LAR and albumin (r = −0.403, *p* ≤ 0.001). There was a moderate positive correlation between FER and the modified Pouchet score (r = 0.396, *p* ≤ 0.001), AST (r = 0.291, *p* ≤ 0.001), WBC (r = 0.266, *p* = 0.001), and neutrophils (r = 0.264, *p* = 0.001), and a high positive correlation with ferritin (r = 0.890, *p* ≤ 0.001). The correlation analysis revealed a modest positive association between CAR and the modified Pouchet score (r = 0.306, *p* = 0.006), a strong positive correlation with CRP (r = 0.969, *p* < 0.001), and a modest negative correlation with albumin (r = −0.259, *p* = 0.001). The SII showed a modest positive correlation with the modified Pouchet score (r = 0.361, *p* = 0.001), a moderate positive correlation with platelets (r = 0.567, *p* = 0.001) and white blood cells (r = 0.702, *p* = 0.001), and a high positive correlation with neutrophils (r = 0.810, *p* < 0.001). There was a statistically significant weak positive association between the NLR and the modified Pouchet score (r = 0.388, *p* ≤ 0.001) and ferritin (r = 0.289, *p* ≤ 0.001). Additionally, there was a statistically significant moderate positive correlation between the WBC count (r = 0.588, *p* ≤ 0.001) and the neutrophil count (r = 0.729, *p* ≤ 0.001). Table 5 shows that there was a modest positive correlation between PLR and modified Pouchet score (r = 0.288, *p* = 0.011), as well as between PLR and platelets (r = 0.421, *p* ≤ 0.001).

## 4. Discussion

In our study, we evaluated the performance of the inflammatory indices LAR, SII, CAR, NLR, PLR, PLR, FER, LAR + FER, LAR + PLR, and LAR + ferritin, which are known to be useful in the diagnosis and prognosis of many diseases, in the diagnosis of AOSD and whether they are at least as successful as ferritin. Although there are some data in the literature on disease activity associated with LAR, NLR, PLR, CAR, SII, and FER in some rheumatologic diseases, there has been no study comparing these indices in AOSD [9,10,12,21,22,23]. In our study, ferritin, ferritin-related FER, FER + LAR, and ferritin + LAR had the highest sensitivity and specificity, while NLR and SII had similar sensitivity and specificity. While CAR was not useful, the sensitivity and specificity of LAR were not superior to the other indices.

The diagnostic process for AOSD is indeed not always as straightforward as desired. In particular, the lack of a specific laboratory marker complicates the diagnostic process. For this reason, some markers with high diagnostic performance for AOSD were investigated. In our study, we tried to find out whether LAR could be a new diagnostic marker by comparing it with other indices. In particular, in the search for an index with sufficiently high sensitivity and specificity, we discussed whether LAR really contributes to clinical practice. The fact that AOSD is a common diagnosis among inflammatory rheumatic diseases when FUO is investigated increases the importance of finding a new marker or index for differentiation with patients in the FUO group.

There was an earlier study comparing inflammatory indices for AOSD. This study compared SII, SII + ferritin, CAR, FER, PNI, and AGR. The study included 164 patients in the AOSD group. According to the AUC results, SII had the best diagnostic ability for AOSD (AUC: 0.859), while CAR, FER, PNI, and AGR had intermediate diagnostic power. In addition, SII had the highest specificity (91.5%) and positive LR [9]. The sensitivity and specificity of ferritin, one of the most important diagnostic tools in AOSD, were reported to be 40.8% and 80%, respectively [24]. In our study, the sensitivity and specificity of the diagnostic performance of serum ferritin for the cutoff value of 3978.0 µg/L to differentiate patients with AOSD from the FUO group were 46.20% and 92.90%, respectively, which were higher. In addition, FER (AUC:0.737) and LAR + ferritin (AUC = 0.729), had the two highest AUC values in our study, while LAR + FER (AUC = 0.696), SII (AUC:0.657), and NLR (AUC:0.609) were the other three indices with high AUC values and showed lower diagnostic performance compared to this study. In a study in which the diagnostic performance was calculated in relation to the ferritin level > 1500 ng/dL in 58 LAR-related AOSD patients without a control group, the AUC was 0.890, sensitivity was 83.70% and specificity 80.0% [10]. In our study, the AUC value of the LAR was 0.581, the sensitivity was 76.90%, and the specificity was 48.60%. In our study, the diagnostic performances of CAR (AUC:0.317) and PLR (AUC:0.477) were also lower than those of SII, FER, LAR, and NLR.

Data on the diagnostic utility of cytokines other than IL-18 are limited. In distinguishing between sepsis and AOSD, the sensitivity and specificity of IL-18 were 88% and 78% for a cutoff value of 148.9 pg/mL in one study and 91.7% and 99.1% for a cutoff value of 366.1 pg/mL in another study [25,26]. Although IL-18 has high sensitivity and specificity, its use is not recommended in clinical practice because it cannot be tested in every laboratory, its cost is high and its testing takes time [9]. Therefore, inflammatory indices can be used to differentiate patients with AOSD from the group of patients who are examined mainly for fever and whose clinical findings are similar to those of AOSD. However, it is uncertain to what extent these indices, which have both specificity and sensitivity values that are similar or slightly different to those of ferritin, will contribute to clinical practice. The fact that these indices, which are recommended in most studies as diagnostic or prognostic in autoimmune diseases, sepsis and solid malignancies, have a high degree of sensitivity and specificity for the performance of these indices in the diagnosis of AOSD and that the index that can achieve both sensitivity and specificity is FER, which also incorporates ferritin in the calculation, once again demonstrates the importance of the classical approach.

Multiple research efforts are now examining the correlation between disease activity and test markers of inflammation in rheumatic diseases. PNI with disease activity in SLE, CAR with disease activity in vasculitis, CAR, SII, and NLR with disease activity in rheumatoid arthritis are some inflammatory indices studied [27,28,29,30]. One study investigated the correlation of SII, SII+ ferritin, CAR, AGR, PNI, and FER indices with Pouchet score, CRP, ferritin, ESR, LDH, total bilirubin, hemoglobin, AST, and ALT in the diagnosis of AOSD. In this investigation, there was a positive correlation seen between SII+ ferritin and all illness indicators, with the exception of hemoglobin. The associations between FER and ferritin, CAR and CRP, SII+ ferritin, and ferritin were weak and moderate, whereas all other relationships were poor [9]. In our study, other indices except LAR showed weak positive correlation with modified Pouchet score, FER and ferritin, SII and platelets, WBC and neutrophil count, NLR and WBC and neutrophil count, while CAR and CRP showed positive correlation.

### Limitations

The limitations of our study are that it was conducted at a single center, retrospective in nature, and had a very small sample size of patients. However, due to the long study period, potential variations in measurement techniques may have influenced the indices and their interpretation. The inclusion of the FUO group as a control, which encompasses a heterogeneous set of diagnoses, including autoinflammatory diseases, is a limitation of this study. This overlap may have introduced confounding effects; however, FUO was chosen due to its clinical similarity to AOSD, which poses significant diagnostic challenges. Future studies with a prospective design are warranted to minimize these effects.

## 5. Conclusions

In our study, we first evaluated LAR as a new inflammation index that could help in the diagnosis of AOSD. Compared to ferritin, LAR had high sensitivity but low specificity. Compared to other indices, LAR had a lower sensitivity than LAR + PLR, LAR + FER, PLR, and CAR and a higher specificity than LAR + PLR, CAR, and PLR. Since the specificity of CAR was very low in our study, it is predicted that it is not useful and indices other than PLR and CAR can be used. However, it should be noted that high sensitivity, specificity, and AUC values were not observed in many studies. Since the LAR has not yet been used in AOSD in a study with similar methodology, the diagnostic value of the LAR and the comparison of the LAR with five different indices and routinely used laboratory values such as ferritin make our study valuable. In addition, the use of patients with FUO with baseline characteristics of AOSD as a control group was another strength of our study.

## Figures and Tables

**Figure 1 diagnostics-14-02780-f001:**
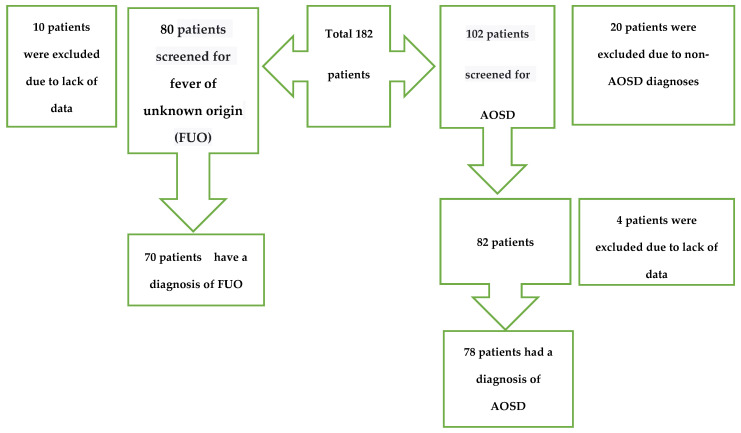
Flowchart for study.

**Figure 2 diagnostics-14-02780-f002:**
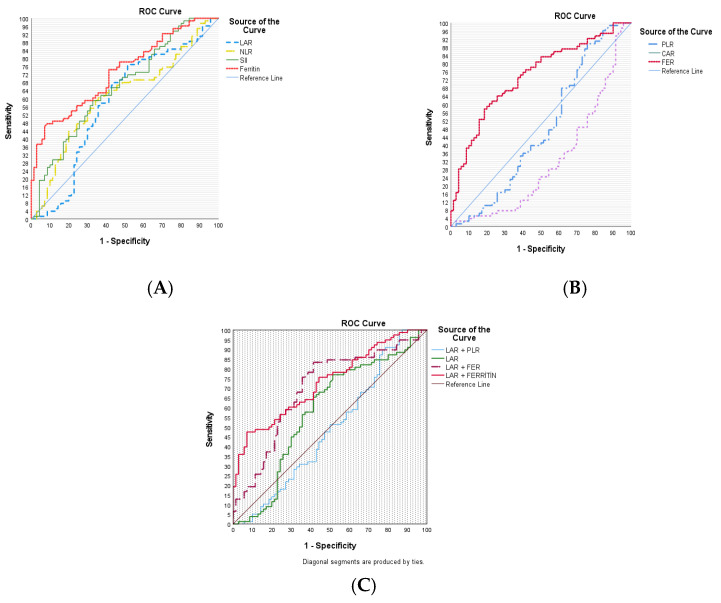
Receiver operating characteristic (ROC) curves pertaining to the diagnostic utility inflammation indices and markers for adult-onset Still’s disease (AOSD) patients. (**A**) For an AOSD diagnosis, the area under the curve (AUC) was 0.581 (95% CI, 0.485–0.676) for LDH-to-albumin ratio (LAR), 0.609 (95% CI, 0.517–0.701) for neutrophil-to-lymphocyte ratio (NLR), 0.657 (95% CI, 0.569–0.744) for Systemic immune-inflammation index (SII), 0.732 (95% CI, 0.653–0.811) for ferritin. (**B**) For an AOSD diagnosis, the AUC was 0.477 (95% CI, 0.381–0.574) for platelet-to-lymphocyte ratio (PLR), 0.317 (95% CI, 0.230–0.404) for C-reactive protein/albumin ratio (CAR), 0.737 (95% CI, 0.657–0.817) for ferritin-to-erythrocyte sedimentation rate ratio (FER). (**C**) For an AOSD diagnosis, the AUC was 0.581 (95% CI, 0.485–0.676) for LAR, 0.488 (95% CI, 0.393–0.584) for LAR + PLR, 0.696 (95% CI, 0.609–0.783) for LAR + FER and, 0.729 (95% CI, 0.649–0.808) for LAR + ferritin.

**Table 1 diagnostics-14-02780-t001:** AOSD and FUO patients characteristics (*n* = 148).

	AOSD Patients (*n* = 78)	FUO Patients (*n* = 70)	*p* Value *
Age (years), mean ± S.D. (min., max., median)	48.82 ± 15.41 (19.0, 82.0, 45.0)	63.39 ± 15.35 (20, 89, 67)	<0.001 †
Gender, female/male (%)	53 /25 (67.90 /32.10)	27 /43 (38.60 /61.40)	<0.001 ‡
Comorbidity, *n* (%)	22 (28.20)	53 (75.70)	<0.001 ‡
DM, *n* (%)	4 (5.10)	18 (25.70)	<0.001 ‡
HT, *n* (%)	12 (15.40)	30 (42.90)	<0.001 ‡
CKD, *n* (%)	2 (2.60)	6 (8.60)	0.107 ‡
CAD, *n* (%)	4 (5.10)	8 (11.40)	0.161 ‡
HF, *n* (%)	2 (2.60)	4 (5.70)	0.332 ‡
HL, *n* (%)	2 (2.60)	4 (5.70)	0.332 ‡
Asthma, *n* (%)	3 (3.80)	2 (2.90)	0.740 ‡
MAS, *n* (%)	10 (12.80)	5 (7.10)	0.253 ‡
Diagnosis of malignancy, *n* (%)	3 (3.85)	37 (52.86)	-
Hematologic malignancy, *n* (%)	3 (3.85)	20 (28.57)	-
Lymphoma, *n* (%)	0	16 (22.86)	-
Hodgkin lymphoma, *n* (%)	0	4 (5.71)	-
Non-Hodgkin lymphoma, *n* (%)	0	12 (17.14)	-
Other hematological malignancy, *n* (%)	3 (3.85)	4 (5.71)	-
Solid malignancy, *n*(%)	0	17 (24.29)	-
Clinical findings			
Fever, *n* (%)	70 (89.70)	70 (100.0)	0.006 ‡
Arthralgia, *n* (%)	71 (91.0)	17 (24.30)	<0.001 ‡
Atrhritis, *n* (%)	61 (78.20)	7 (10.0)	<0.001 ‡
Monoarthritis, *n* (%)	10 (12.80)	4 (5.70)	
Oligoarthritis, *n* (%)	28 (35.90)	0	<0.001 ‡
Polyarthritis, *n* (%)	23 (29.50)	3 (4.30)	
Myalgia, *n* (%)	27 (34.62)	35 (50.0)	
Typical nonpruritic salmon-pink skin rash, *n* (%)	58 (74.40)	3 (4.30)	0.058 ‡
Sore throat, *n* (%)	36 (46.16)	3 (4.30)	<0.001 ‡
Pleuritis, *n* (%)	4 (5.10)	12 (17.10)	<0.001 ‡
Pericarditis, *n* (%)	2 (2.60)	8 (11.40)	0.019 ‡
LAP, *n* (%)	17 (21.80)	37 (52.90)	0.032 ‡
HM, *n* (%)	12 (15.40)	23 (32.90)	<0.001 ‡
SM, *n* (%)	13 (16.70)	19 (27.10)	0.013 ‡
Laboratory findings			
ANA negativity, *n* (%)	62 (79.50)	26 /39 (66.67)	-
RF negativity, *n* (%)	73 (93.60)	25 /29 (86.20)	-
Anti-CCP negativity, *n* (%)	77 (98.70)	22 /24 (91.67)	-
AST (U/L), median (P25, P75)	22.0 (16.0, 44.50)	29.50 (18.0, 58.25)	0.165 †
ALT (U/L), median (P25, P75)	22.0 (16, 59.75)	25.0 (14.0, 44.0)	0.956 †
GGT (U/L), median (P25, P75)	67.0 (34.0, 102.25)	54.0 (31.0, 144.0)	0.660 †
ALP (U/L), median (P25, P75)	89.0 (66.75, 125.25)	19.50 (80.50, 261.0)	0.003 †
LDH (U/L), median (P25, P75)	363.50 (277.0, 483.75)	229.50 (166.50, 394.0)	<0.001 †
Albumin (g/dL), mean ± S.D. (min., max., median)	3.76 ±0.66 (2.10, 5.20, 3.85)	3.92 ± 5.58 (2.50, 3.50, 3.0)	<0.001 §
WBC (10*9/L), median (P25, P75)	12.39 (8.62, 19.82)	7.56 (5.32, 11.30)	<0.001 †
Neutrophil (10*9/L), median (P25, P75)	9.19 (5.80, 16.03)	5.14 (3.36, 7.81)	<0.001 †
Neutrophil ratio (> 80%), *n* (%)	38 (48.70)	12 (17.10)	<0.001 ‡
Lymphocyte (10*9/L), median (P25, P75)	1.73 (1.18, 2.41)	1.27 (0.80, 1.89)	0.002 †
Hemoglobin (g/dL), median (P25, P75)	12.15 (10.60, 13.90)	9.35 (8.50, 10.79)	<0.001 †
Platelet (10*9/L), median (P25, P75)	293.50 (238.17, 363.25)	246.50 (127.06, 337.78)	0.006 †
CRP (mg/L), median (P25, P75)	10.30 (4.24, 28.25)	43.50 (9.43, 120.25)	<0.001 †
ESR (mm/h), median (P25, P75)	54.50 (34.25, 85.25)	50.50 (28.75, 77.25)	0.461 §
ferritin (µg/L), median (P25, P75)	2302.0 (625.25, 11281.25)	580.50 (230.50, 1652.25)	<0.001 †
Pouchet score, median (P25, P75)	5 (4, 6)	-	-
Inflammation indices			
LAR, median (P25, P75)	101.09 (72.58, 142.53)	75.95 (54.50, 131.95)	0.090 †
SII [(P × N)/L)], median (P25, P75)	1614.32 (656.22, 3283.61)	955.25 (473.28 (1923.38)	<0.001 †
CAR, median (P25, P75)	3.32 (1.19, 8.50)	9.55 (3.25, 44.63)	<0.001 †
NLR, median (P25, P75)	5.98 (2.52, 11.24)	3.80 (2.35, 6.55)	0.022 †
PLR, median (P25, P75)	165.58 (122.36, 252.74)	193.95 (101.15, 319.35)	0.634 †
FER, median (P25, P75)	64.28 (15.98, 203.14)	12.73 (5.45, 29.10)	<0.001 †
LAR + PLR ( P25, P75)	276.23 (205.92, 385.70)	283.85 (202.40, 413.80)	0.806 †
LAR + FER (P25, P75)	192.55 (110.15, 323.46)	85.65 (64.01, 164.54)	<0.001 †
LAR + ferritin (P25, P75)	2420.0 (733.63, 11423.99)	652.40 (308.53, 1718.15)	<0.001 †

AOSD: adult-onset Still’s disease, FUO: fever of unknown origin, P25: 25th percentile; P75: 75th percentile; S. D.: standard deviation; Min.: minimum; Max: maximum; DM: diabetes mellitus; HT: hypertension; CKD: chronic kidney disease; CAD: coronary artery disease; HF: heart failure; HL: hyperlipidemia; AML: acute myeloid leukemia; MDS: myelodysplastic syndrome; MAS: macrophage activation syndrome; LAP: lymphadenopathy; HM: hepatomegaly; SM: splenomegaly; ANA: antinuclear antibody; RF: rheumatoid factor; Anti-CCP: anti-cyclic citrullinated peptide antibodies; ALT: alanine aminotransferase; AST: aspartate aminotransferase; GGT: gamma-glutamyl transpeptidase; ALP: alkaline phosphatase; WBC: white blood cells; CRP: C-reactive protein; ESR: erythrocyte sedimentation rate (1st hour); LDL: low-density lipoprotein; HDL: high-density lipoprotein; LDH: lactate dehydrogenase; LAR: LDH-to-albumin ratio, FER: ferritin-to-erythrocyte sedimentation rate ratio, CAR: C-reactive protein/albumin ratio, SII: Systemic immune-inflammation index (platelet count × neutrophil/lymphocyte count at diagnosis), NLR: neutrophil-to-lymphocyte ratio, PLR: platelet-to-lymphocyte ratio, *****: *p* values < 0.05 were considered statistically significant, †: Mann–Whitney U test, ‡: chi-square test, §: independent samples t test.

**Table 2 diagnostics-14-02780-t002:** Treatment and disease characteristics of AOSD patients (*n* = 78).

Variables	Data
Disease duration (months), median (P25, P75)	132.0 (58.0, 160.0)
Duration from symptom onset to diagnosis (months), median (P25, P75)	0 (0, 1)
Disease patterns	
Monocyclic systemic/self-limited, *n* (%)	37 (47.40)
Polycyclic systemic/intermittent, *n* (%)	27 (37.60)
Chronic articular, *n* (%)	14 (17.90)
Treatment	
NSAID, *n* (%)	44 (56.40)
GC	
Oral GC, *n* (%)	44 (56.40)
Pulse GC, *n* (%)	34 (43.60)
cDMARDs	
HCQ, *n*(%)	48 (61.50)
Sulfasalazin, *n* (%)	3 (3.80)
Methotrexate, *n* (%)	66 (84.60)
Leflunomide, *n* (%)	16 (20.50)
IL-1 inhibitors	
Anakinra, *n* (%)	14 (17.90)
Canakinumab, *n* (%)	5 (6.40)
TNFα inhibitors	
Adalimumab, *n*(%)	3 (3.80)
Certolizumab pegol, *n* (%)	4 (5.10)
Etanercept	5 (6.40)
Tocilizumab, *n* (%)	14 (17.90)
Tofacitinib, *n* (%)	4 (5.10)
Baricitinib, *n* (%)	1 (1.30)

AOSD: adult-onset Still’s disease; P25: 25th percentile; P75: 75th percentile; NSAID: non-steroidal anti-inflammatory drugs; GC: glucocorticoid; c-DMARDs: conventional disease-modifying antirheumatic drugs; HCQ: hydroxychloroquine; IL-1: interleukin-1; TNF: tumor necrosis factor.

**Table 3 diagnostics-14-02780-t003:** ROC analysis results of the inflammation indices and ferritin level of the patient group diagnosed with AOSD and the FUO group.

Variables	AUC	95% CI	*p* Value *	Cutoff	Sensitivity (%)	Specificity (%)	+LR	−LR
LAR	0.581	0.485–0.676	0.090	70.98	76.90	48.60	1.50	0.48
NLR	0.609	0.517–0.701	0.022	5.30	59.0	65.70	1.72	0.62
SII	0.657	0.569–0.744	0.045	1356.34	59.0	67.10	1.79	0.61
CAR	0.317	0.230–0.404	0.000	0.23	100.0	4.3	1.04	0
FER	0.737	0.657–0.817	0.000	39.38	57.70	81.40	3.11	0.52
PLR	0.477	0.381–0.574	0.634	103.89	87.20	25.70	1.17	0.50
LAR + PLR	0.488	0.393–0.584	0.806	154.93	98.70	14.30	1.15	0.08
LAR + FER	0.696	0.609–0.783	0.044	95.49	83.0	58.60	1.41	0.20
ferritin	0.732	0.653–0.811	0.000	3978.0	46.20	92.90	6.46	0.58
LAR + ferritin	0.729	0.649–0.808	0.000	3974.75	47.40	92.90	6.64	0.57

LAR: LDH/albumin ratio; NLR: neutrophil/lymphocyte ratio; SII: systemic immune-inflammation index (platelet count × neutrophil/lymphocyte count at diagnosis). CAR: CRP/albumin ratio; PLR: platelet/lymphocyte ratio; FER: ferritin/ESR ratio; OR: odds ratio; CI: confidence interval; ROC: receiver operating characteristic. ESR: erythrocyte sedimentation rate (1st hour), *: *p* values < 0.05 were considered statistically significant.

**Table 4 diagnostics-14-02780-t004:** Comparison of the inflammation indices and ferritin cutoffs determined using the ROC curve in the AOSD group and the FUO group.

Variables	AOSD Group (*n* = 78)	FUO Group (*n* = 70) (*n*, %)	*p*-Value *
LAR (>70.98)	60 (76.92)	36 (51.43)	0.001 †
SII (>1356.34)	46 (58.97)	23 (32.86)	0.001 †
NLR (>5.30)	46 (58.97)	24 (34.29)	0.003 †
CAR (>0.23)	78 (100.0)	68 (97.14)	0.133 †
FER (>39.38)	44 (56.41)	14 (20.0)	<0.001 †
PLR (>103.89)	68 (87.18)	52 (74.29)	0.046 †
LAR + PLR (>154.93)	77 (98.72)	63 (90.0)	0.019 †
LAR + FER (>95.49)	65 (83.33)	29 (41.43)	<0.001 †
LAR + ferritin (>3974.75)	37 (47.44)	5 (7.14)	<0.001 †

LAR: LDH-to-albumin ratio, FER: ferritin-to-erythrocyte sedimentation rate ratio, CAR: C-reactive protein/albumin ratio, SII: Systemic immune-inflammation index (platelet count × neutrophil/lymphocyte count at diagnosis), NLR: neutrophil-to-lymphocyte ratio, PLR: platelet-to-lymphocyte ratio, AOSD: adult-onset Still’s disease, FUO: fever of unknown origin, ROC: receiver operating characteristic *****: *p* values < 0.05 were considered statistically significant. †: chi-square test.

**Table 5 diagnostics-14-02780-t005:** Correlation among disease activity indicators and laboratory inflammatory markers in AOSD patients.

	Correlation Coefficient, r (*p* Value *)					
Disease Activity Indicators	LAR	FER	CAR	SII	NLR	PLR
Modified Pouchet score, median (P25, P75)	0.220 (0.053)	0.396 (<0.001)	0.306 (0.006)	0.361 (0.001)	0.388 (<0.001)	0.288 (0.011)
ferritin (µg/L), median (P25, P75)	0.461 (<0.001)	0.890 (<0.001)	0.120 (0.146)	0.207 (0.011)	0.289 (<0.001)	−0.009 (0.917)
AST (U/L), median (P25, P75)	0.367 (<0.001)	0.291 (<0.001)	0.206 (0.012)	0.073 (0.378)	0.244 (0.003)	0.091 (0.273)
ALT (U/L), median (P25, P75)	0.205 (0.012)	0.222 (0.007)	0.064 (0.436)	0.074 (0.373)	0.175 (0.033)	0.003 (0.968)
Hemoglobin (g/dL), median (P25, P75)	−0.069 (0.405)	0.141 (0.087)	−0.226 (0.006)	−0.052 (0.537)	−0.029 (0.729)	−0.237 (0.004)
Platelet (10*9/L), median (P25, P75)	−0.124 (0.133)	−0.145 (0.078)	−0.075 (0.365)	0.567 (0.001)	0.160 (0.052)	0.421 (<0.001)
WBC (10*9/L), median (P25, P75)	0.181 (0.027)	0.266 (0.001)	0.068 (0.412)	0.702 (<0.001)	0.588 (<0.001)	0.084 (0.313)
Neutrophil (10*9/L), median (P25, P75)	0.224 (0.006)	0.264 (0.001)	0.088 (0.290)	0.810 (<0.001)	0.729 (<0.001)	0.229 (0.005)
Albumin (g/dL), mean±S.D	−0.413 (<0.001)	−0.027 (0.747)	−0.259 (0.001)	0.006 (0.940)	−0.089 (0.280)	−0.162 (0.050)
ESR (mm/h), median (P25, P75)	−0.080 (0.333)	−0.215 (0.009)	0.145 (0.078)	0.203 (0.013)	0.092 (0.267)	0.218 (0.008)
CRP (mg/L), median (P25, P75)	−0.035 (0.679)	−0.003 (0.967)	0.969 (<0.001)	0.128 (0.121)	0.204 (0.013)	0.135 (0.102)

P25: 25th percentile; P75: 75th percentile LAR: LDH-to-albumin ratio, FER: ferritin-to-erythrocyte sedimentation rate ratio, CAR: C-reactive protein/albumin ratio, SII: systemic immune-inflammation index (platelet count × neutrophil/lymphocyte count at diagnosis), NLR: neutrophil-to-lymphocyte ratio, PLR: platelet-to-lymphocyte ratio; CRP: C-reactive protein; ESR: erythrocyte sedimentation rate; WBC: white blood cells; ALT: alanine aminotransferase; AST: aspartate aminotransferase; AOSD: adult-onset Still’s disease, *: *p* values < 0.05 were considered statistically significant.

## Data Availability

Data are available upon reasonable request to the corresponding author.

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
