# Peer review of "Does the LDH/Albumin Ratio Bring Novelty? A Comparative Analysis with Inflammatory Indices and Combined Models in Adult-Onset Still’s Disease"

_diagnostics, 2024, doi:10.3390/diagnostics14242780_

Round 1
Reviewer 1 Report
Comments and Suggestions for Authors
The authors aimed to to assess the effectiveness of LAR in comparison to other immune inflammatory indices (NLR, PLR, SII, CAR, and FER) in patients with AOSD who had Fever of Unknown Origin (FUO) as a control group. The study has some inherent limitations.
The title of the study is not in concordance with the conclusion of the study. The long period of study necessitates review of techniques for measuring parameters. The evolution of methods have a bearing on the calculated indices and the subsequent interpretation.
Proof reading required to remove sentences such as “A few sentences to place the question addressed 19 in a broader context and highlight the purpose of the study” from the abstract.
The overlap of autoinflammatory disease in FUO group will act as a confounder of the study.
Author Response
Comments 1: The authors aimed to to assess the effectiveness of LAR in comparison to other immune inflammatory indices (NLR, PLR, SII, CAR, and FER) in patients with AOSD who had Fever of Unknown Origin (FUO) as a control group. The study has some inherent limitations. The title of the study is not in concordance with the conclusion of the study.
Answer 1:
Dear Reviewer,
Thank you for your thoughtful feedback and valuable suggestions. We appreciate the opportunity to address your concerns and clarify our manuscript. Below are our detailed responses to the issues raised:
We agree that the title could better reflect the conclusions drawn from the study. To address this, we propose revising the title to: ‘Does the LDH/Albumin Ratio Bring Novelty? A Comparative Analysis with Inflammatory Indices and Combined Models in Adult-Onset Still’s Disease’
Comments 2: The long period of study necessitates review of techniques for measuring parameters. The evolution of methods have a bearing on the calculated indices and the subsequent interpretation.
Answer 2: We acknowledge this limitation inherent to retrospective studies. Although consistent protocols were applied during data collection, potential variations in measurement techniques over the study period may have influenced the results. We will include the following statement in the limitations section: ‘However, due to the long study period, potential variations in measurement techniques may have influenced the indices and their interpretation.’
Comments 3: Proof reading required to remove sentences such as “A few sentences to place the question addressed 19 in a broader context and highlight the purpose of the study” from the abstract.
Answer 3: We appreciate this observation. The mentioned sentence was inadvertently retained from an earlier draft. We have removed it and carefully proofread the manuscript to ensure that the abstract is concise and free of irrelevant content.
Comments 4: The overlap of autoinflammatory disease in FUO group will act as a confounder of the study.
Answer 4: We recognize this limitation and agree that the heterogeneous nature of the FUO group, including potential overlap with autoinflammatory diseases, could introduce confounding effects. However, FUO was chosen as a control group due to its clinical similarity to AOSD, which poses diagnostic challenges. We have added the following to the limitations section:
‘The inclusion of the FUO group as a control, which encompasses a heterogeneous set of diagnoses, including autoinflammatory diseases, is a limitation of this study. This overlap may have introduced confounding effects; however, FUO was chosen due to its clinical similarity to AOSD, which poses significant diagnostic challenges. Future studies with a prospective design are warranted to minimize these effects.’
Reviewer 2 Report
Comments and Suggestions for Authors
The authors proposed that lactate dehydrogenase to albumin ratio (LAR) used in the clinics might not be an accurate diagnostic criteria to differentiate AOSD and FUO. Through a retrospective evaluation, the authors found ferritin/erythrocyte sedimentation rate (FER) and CRP/albumin ratio (CAR) showing better performance than LAR.
There are a few suggestions that might help improve the manuscripts:
1. Line 84-85 “in patients with AOSD who had Fever of Unknown Origin (FUO) as a control group” and methods:
a. Was the comparison done between the patients with AOSD +/- FUO? Then in Figure 1 showed the patients were split between having AOSD or FUO. Please address the reason of using FUO to stratify the patients into two groups, and revise if it is correct as it’s confusing in the abstract: “The objective of this study was to assess the diagnostic accuracy of the lactate dehydrogenase to albumin ratio (LAR) in Adult Onset Still's Disease (AOSD) in comparison to patients with fever of unknown origin (FUO).”
2. Will combinations of these inflammatory indices be better in the diagnosis and prognosis of AOSD vs. FUO? would be a good addition of the analysis to include at least some of those indices and compare with single indices.
Author Response
Comments 1: The authors proposed that lactate dehydrogenase to albumin ratio (LAR) used in the clinics might not be an accurate diagnostic criteria to differentiate AOSD and FUO. Through a retrospective evaluation, the authors found ferritin/erythrocyte sedimentation rate (FER) and CRP/albumin ratio (CAR) showing better performance than LAR.
There are a few suggestions that might help improve the manuscripts:
- Line 84-85 “in patients with AOSD who had Fever of Unknown Origin (FUO) as a control group” and methods:
- Was the comparison done between the patients with AOSD +/- FUO? Then in Figure 1 showed the patients were split between having AOSD or FUO. Please address the reason of using FUO to stratify the patients into two groups, and revise if it is correct as it’s confusing in the abstract: “The objective of this study was to assess the diagnostic accuracy of the lactate dehydrogenase to albumin ratio (LAR) in Adult Onset Still's Disease (AOSD) in comparison to patients with fever of unknown origin (FUO).”
Answer 1: Dear Reviewer,
Thank you for your constructive comments and valuable suggestions. We have carefully considered your feedback to improve the clarity and scientific rigor of our manuscript.
The FUO group was chosen as a control group to represent patients with similar clinical features to AOSD but with different underlying pathologies. This design allowed us to evaluate the diagnostic accuracy of LAR and other indices in differentiating AOSD from FUO. The groups were clearly divided into two distinct populations: AOSD and FUO. We will revise the manuscript to clarify this point as follows:
Abstract: ..’ The objective of this study was to evaluate the diagnostic accuracy of the lactate dehydrogenase to albumin ratio (LAR) in Adult-Onset Still's Disease (AOSD) and compare it with other inflammatory indices, using patients with Fever of Unknown Origin (FUO) as a control group due to their overlapping clinical features with AOSD.’…
Introduction: ..’ The objective of this study was to evaluate the diagnostic accuracy of the lactate dehydrogenase to albumin ratio (LAR) in Adult-Onset Still's Disease (AOSD) and compare it with other inflam-matory indices, using patients with Fever of Unknown Origin (FUO) as a control group due to their overlapping clinical features with AOSD.
‘..
We will also revise Tables and Figures and its legend to ensure that the grouping process and rationale are clear.
Comments 2: Will combinations of these inflammatory indices be better in the diagnosis and prognosis of AOSD vs. FUO? would be a good addition of the analysis to include at least some of those indices and compare with single indices.
Answer 2: We appreciate this insightful suggestion. Following your recommendation, we evaluated combinations of LAR with FER, PLR, and Ferritin. The results showed that these combinations significantly improved diagnostic performance compared to LAR alone. Below are the changes we have made to the article:
Abstract:
Background/Objectives: ..'The study also compared LAR's diagnostic performance with other inflammatory indices like the serum im-mune-inflammatory index (SII), ferritin/erythrocyte sedimentation rate (FER), CRP/albumin ratio (CAR), platelet/lymphocyte ratio (PLR), and neutrophil/lymphocyte ratio (NLR), as well as its combinations with FER, PLR, and ferritin (LAR+FER, LAR+PLR, LAR+Ferritin)..' Results: ..'ROC analysis showed significant p-values (p<0.05) for indices other than LAR (p = 0.090) LAR+PLR (p=0.806), and PLR (p = 0.634) in diagnosing AOSD. The highest specificity was found in LAR+Ferritin (92.90 %), and the highest sensitivity in CAR (100.0%)...' NLR, SII, FER and LAR+FER were the indices with both sensitivity and specificity above 50%. LAR had a sensitivity of 76.90% and a specificity of 48.60%. The cut-off values were 3978.0 µg/L for ferritin and 70.98 for LAR.
Introduction:
..’ The study also compared LAR's diagnostic per-formance with other inflammatory indices like the serum immune-inflammatory index (SII), ferri-tin/erythrocyte sedimentation rate (FER), CRP/albumin ratio (CAR), platelet/lymphocyte ratio (PLR), and neutrophil/lymphocyte ratio (NLR), as well as its combinations with FER, PLR, and ferritin (LAR+FER, LAR+PLR, LAR+Ferritin).‘..
Material and Methods:
Variables and Procedures: …’ LAR, NLR, PLR, CAR, FER, SII, LAR+FER, LAR+PLR and LAR+Ferritin were calculated and recorded. The reason for combining LAR only with PLR, FER, and ferritin was their high specificity and sensitivity rates’..
3.Results
3.1. Clinical and laboratory features, treatment and follow-up of AOSD patients
..’With the exception of LAR (p=0.090), PLR (p=0.634) and LAR+PLR (p=0.806), the other inflammatory indices differed statistically significantly between the two groups. While CAR, PLR and LAR+PLR were lower in the AOSD group, SII, NLR, FER and LAR were higher.’..
3.2. Relationship between inflammatory indices and the diagnosis of AOSD
..’The results of the combinations LAR+PLR, LAR+FER, and LAR+Ferritin were as follows, respectively: 0.488 (95% CI, 0.393- 0.584), 0.696 (95% CI, 0.609- 0.783) and 0.729 (95% CI, 0.649- 0.808). .’
3.2. Relationship between inflammatory indices and the diagnosis of AOSD
..’In the ROC analysis, the p-values for SII, NLR, FER and CAR were statistically significant, except for LAR (p=0.090), PLR (p=0.634), and LAR+PLR (p=0.806). The index with the highest specificity were FER (81.40%) and LAR+Ferritin (92.90%), the index with the highest sensitivity were CAR (100.0%) and LAR+PLR (p=98.70%).’..
Discussion:
…’In addition, FER (AUC:0.737) and LAR+Ferritin (AUC=0.729), had the two highest AUC value in our study, while LAR+FER (AUC=0.696), SII (AUC:0.657), and NLR (AUC:0.609), and were the other three indices with high AUC values and showed lower diagnostic performance compared to this study.’..
**Non-AOSD group was changed to FUO group.
**FER+LAR, Ferritin+LAR and PLR+LAR were added to the relevant tables. ROC analysis was performed. sensitivity, specificity, positive LR and negative LR were calculated. Option C was added to Figure 2.
Round 2
Reviewer 1 Report
Comments and Suggestions for Authors
All comments have been addressed.